# Stretchable Thermochromic Fluorescent Fibers Based on Self-Crystallinity Phase Change for Smart Wearable Displays

**DOI:** 10.3390/polym16243575

**Published:** 2024-12-21

**Authors:** Yongmei Guo, Zixi Hu, Luyao Zhan, Yongkun Liu, Luping Sun, Ying Ma

**Affiliations:** 1Fujian Key Laboratory of Novel Functional Textile Fibers and Materials, Clothing and Design Faculty, Minjiang University, Fuzhou 350108, China; 2Engineering Research Center of Technical Textiles, Ministry of Education, College of Textiles, Donghua University, Shanghai 201620, China; 2220304@mail.dhu.edu.cn (Z.H.); zzly1777@163.com (L.Z.); 13337218295@163.com (Y.L.); lupingsun314@163.com (L.S.)

**Keywords:** smart fiber, stretchable, textile display, electrochromic fluorescent emission, self-crystallinity phase change

## Abstract

Smart fibers with tunable luminescence properties, as a new form of visual output, present the potential to revolutionize personal living habits in the future and are receiving more and more attention. However, a huge challenge of smart fibers as wearable materials is their stretching capability for seamless integration with the human body. Herein, stretchable thermochromic fluorescent fibers are prepared based on self-crystallinity phase change, using elastic polyurethane (PU) as the fiber matrix, to meet the dynamic requirements of the human body. The switching fluorescence-emitting characteristic of the fibers is derived from the reversible conversion of the dispersion/aggregation state of the fluorophore coumarin 6 (C6) and the quencher methylene blue (MB) in the phase-change material hexadecanoic acid (HcA) during heating/cooling processes. Considering the important role of phase-change materials, thermochromic fluorescent dye is encapsuled in the solid state via the piercing–solidifying method to avoid the dissolution of HcA by the organic solvent of the PU spinning solution and maintain excellent thermochromic behavior in the fibers. The fibers obtained by wet spinning exhibit good fluorescent emission contrast and reversibility, as well as high elasticity of 800% elongation. This work presents a strategy for constructing stretchable smart luminescence fibers for human–machine interaction and communications.

## 1. Introduction

Smart wearable materials have attracted growing attention in recent years due to their great application potential in image display, health monitoring, human–machine interaction, and other aspects, with the development of materials science and technology [1,2,3,4,5,6]. Smart fibers possessing good comfort and excellent adaptability, as the basic building unit of human clothing, have become one kind of important smart wearable material [7,8,9,10,11,12]. Various types of smart fibers with sensing, luminescence, thermal regulation, and energy management capabilities, among others, have been widely investigated for endowing common textiles with various functions besides the conventional warming function. These smart fibers and textiles are anticipated to revolutionize personal living habits, playing an essential and active role in our daily lives. As one type of smart fiber, responsive fibers with tunable luminescence properties provide a new visual output form, increasing the environmental cognition and adaptative abilities of humans [13,14,15,16,17,18]. Considering of the natural advantages of smart luminescent fibers, such as good flexibility, light weight, small diameter, large aspect ratio, implantability, and long signal transmission distance, lots of research has been carried out on their structural improvement strategy and practical application fields. For example, Peng et al. achieved stable luminescence in bendable fabric by using elastic polymer conductive fibers [19]. Wang et al. realized the wireless visual–digital interactions of luminescent fiber through harvesting ambient electromagnetic energy without the need for extra chips or batteries [20]. However, for wearable materials, a huge challenge is their seamless integration with the human body, so it is particularly important to design smart fibers with a stretchable capability to adapt to the dynamic movements of the human body.

Thermochromic fluorescent materials based on self-crystallinity phase change are one of promising responsive luminescent materials due to their features of fast response speed, high emission contrast, and good reversibility [21,22,23]. Specifically, the tunable luminescence behavior derives from the reversible conversion of the aggregation/dispersion states of fluorophores and the corresponding changes in their fluorescent emission during the crystallizing/melting processes of phase-change materials as the matrix of fluorophores. Thermochromic fluorescent materials based on self-crystallinity phase change possess a unique advantage of indifference to fluorophores, and their full-color emission can be achieved according to primary color additive theory. These features are conducive to their various applications in smart display. In addition, thermochromic fluorescence materials based on self-crystallinity phase change can be used in fibers for smart textile display. As demonstrated in our previous works, smart fibers prepared by thermochromic fluorescent materials based on self-crystallinity phase change presented excellent luminescent performances and controllable color-changing temperatures and were suitable for personalized designs for various display applications [24,25]. Despite these efforts, it remains a significant challenge for smart fluorescent fibers based on self-crystallinity phase change to have stretching capability for seamless integration with the human body.

Polyurethane (PU), as an organic polymer produced by the reaction of isocyanates with polyols, has attracted increasing interest due to its satisfactory toughness and elasticity [26,27,28,29,30,31]. Through environmentally friendly wet spinning, PU elastic fibers can be extruded and solidified in water [32,33,34,35]. Herein, we focused on the preparation of stretchable thermochromic fluorescent fibers based on self-crystallinity phase change, using the elastic polymer material PU as the fiber matrix, for meeting the dynamic requirements of the human body. The switching fluorescence-emitting characteristic of the fibers is derived from the reversible conversion of the dispersion/aggregation state of the fluorophore coumarin 6 (C6) and the quencher methylene blue (MB) in the phase-change material hexadecanoic acid (HcA) during heating/cooling processes. Considering the important role of phase-change materials, the thermochromic fluorescent dye including C6, MB, and HcA was encapsuled by calcium alginate via the piercing–solidifying method, avoiding the dissolution of HcA by the solvent DMAc of the PU spinning solution and maintaining excellent thermochromic behavior in the fibers. The fibers obtained by wet spinning exhibited good fluorescent emission contrast and reversibility, as well as high elasticity of 800% elongation.

## 2. Materials and Methods

### 2.1. Materials

Coumarin 6 (C6), methylene blue (MB), hexadecanoic acid (HcA), sodium alginate (SA), and anhydrous calcium chloride (CaCl_2_) were purchased from Shanghai Aladdin Biochemical Technology Co., Ltd. (Shanghai, China). Polyurethane (PU) was obtained from Huntsman Polyurethanes Co., Ltd. (Shanghai, China). N, N-Dimethylacetamide (DMAc) was acquired from Shanghai Adamas Reagent Co., Ltd. (Shanghai, China). Ethanol absolute (EA) was purchased from Sinopharm Chemical Reagent Co., Ltd. (Shang, China). All chemicals were used as received without further purification. Deionized water was used for all the experiments.

### 2.2. Preparation of Thermochromic Fluorescent Capsules

By blending 1.0 μmol/g of C6 and 0.2 μmol/g of the fluorescent quencher MB within the phase-change material HcA, a green thermochromic fluorescent dye was obtained. After undergoing melting and cooling processes, the solid thermochromic fluorescent dye was ground into a powder with a particle size of ≤100 μm for the preparation of thermochromic fluorescent capsules via the piercing–solidifying method. The aqueous solution, consisting of 1.0 wt% dye powder and 1.5 wt% SA, was fed into a 24 G needle (inner diameter of 0.34 mm) and was injected into 5.0 wt% CaCl_2_ solution at a fast rate, forming calcium alginate thermochromic fluorescent capsules. The capsules were dried on a hot plate at 30 °C for 12 h. After drying, the capsules were ground using a grinding bowl to separate any adherent ones and subsequently sieved through a 45-mesh sieve to obtain capsules <400 μm. Finally, the capsules were rinsed several times with EA.

### 2.3. Wet Spinning of Stretchable Thermochromic Fluorescent Fibers

A 10.0 g solution of 20 wt% PU was prepared, and 0.4 g of calcium alginate thermochromic fluorescent capsules was added to it. The mixture was magnetically stirred until the capsules were uniformly dispersed, resulting in a spinning solution. The spinning solution was then fed into a 12 G spinneret (with an inner diameter of 2.3 mm) through a syringe pump at a flow rate of 1 mL/min. Simultaneously, the solution was drawn into a deionized water coagulation bath at a rate 1.5 times faster than the feed rate. After good fiber formation was achieved, the fibers were removed from the coagulation bath and left to dry naturally at room temperature.

### 2.4. Characterization

The morphologies and microstructures of fibers were observed by a field-emission scanning electron microscope (FESEM) (SU8010, Hitachi., Japan). The mechanical properties of fibers were measured using a single-fiber (nanomembrane) strength tester (ST200C, Suzhou, China). Fluorescent photographs were captured under UV irradiation.

## 3. Results and Discussion

### 3.1. Preparation and Characterization

Stretchable thermochromic fluorescent fibers based on self-crystallinity phase change were designed and prepared using the elastic polymer material polyurethane (PU) as the fiber matrix. Common solvents of PU spinning solution are organic N, N-dimethylformamide (DMF) and N, N-dimethylacetamide (DMAc), which could dissolve the phase-change materials of thermochromic fluorescent dyes based on self-crystallinity phase change and destroy the reversible conversion of fluorescent properties. In order to prepare stretchable thermochromic fluorescent PU fibers, the thermochromic fluorescent dye was first capsuled by calcium alginate via the piercing–solidifying method, as shown in Figure 1. The green fluorophore coumarin 6 (C6) was blended with the fluorescent quencher methylene blue (MB) into the phase-change material hexadecanoic acid (HcA) to obtain a green thermochromic fluorescent dye with off–on switching and high reversibility during heating/cooling processes. Microencapsulation is an advanced encapsulation technology that involves encapsulating tiny core materials, such as solids, liquids, or gases, within natural or synthetic polymer walls through physico-chemical means [36,37,38,39]. This process results in the formation of minute and enclosed particles. These formed walls effectively isolate the core material from the external environment, thereby protecting it from external factors and significantly enhancing its stability.

The core principles of the piercing–solidifying method, as a widely used microcapsule preparation technology, lie in the use of sharp-hole devices (e.g., syringes) to precisely drop the mixture of core and wall materials into a solidifying bath, which induces the mixture to rapidly solidify in the solidifying bath and then form microcapsules [40,41,42]. This method is not only simple and easy to operate but also relatively low-cost. It is also worth mentioning that the microcapsules prepared by this method show high homogeneity in morphology and particle size, which helps to ensure the consistency of product quality. As shown in Figure 1, the piercing–solidifying method was utilized to prepare thermochromic fluorescent capsules of calcium alginate. The alginate solution containing the thermochromic fluorescent powders was injected into a highly concentrated calcium chloride solution through a syringe, so that it was rapidly dispersed into tiny droplets in the calcium chloride solution. Subsequently, the calcium ions in the calcium chloride solution rapidly penetrated the alginate droplets and cross-linked them to a gel ball. After drying, the calcium alginate thermochromic fluorescent capsules were successfully produced. As a microencapsulated wall material, alginate exhibits unique advantages, including excellent film formation, ease of handling, and excellent film strength [43,44,45]. More importantly, the calcium alginate produced by the solidification of alginate in CaCl_2_ solution has excellent stability and is insoluble in most organic solvents.

The prepared thermochromic fluorescent capsules are shown in Figure 2a. The wet gel particles were transparent and mostly spherical or ellipsoidal. The diameter of the dry capsules decreased after 12 h of drying and water loss, and some of them appeared to be adherent and of different sizes. Therefore, it was necessary to grind the dried capsules with a grinding bowl, so that the adherent capsules could be separated, and then sieve them through a 45-mesh screen, so as to obtain capsule particles with particle sizes less than 400 μm. In order to eliminate powders that were either unsuccessfully encapsulated or had lost their encapsulation due to excessive grinding, it was also necessary to rinse the obtained capsules several times with ethanol absolute. The capsules, after being sieved and rinsed, are displayed in Figure 2b, exhibiting a more uniform particle size. Meanwhile, the surface morphology of the sieved thermochromic fluorescent capsules was observed using scanning electron microscope (SEM) images, and the result is shown in Figure 2c. The surface of the capsule was uneven and showed irregular granularity due to the presence of the internal thermochromic fluorescent powder and the uneven shrinkage of the SA matrix during water evaporation. As shown in Figure 2d, the particle size of 61 capsules was measured and statistical analysis was performed to obtain the particle size distribution of the capsules. Because the capsules were not regularly spherical, their particle size was calculated as half of the sum of the long and short axes. The particle size of the capsules was distributed in the range of 250~400 μm, with an average diameter of 318 μm.

The thermal luminescence performance of the prepared capsules, as presented in Figure 2, exhibited reversible off–on switching of bright green fluorescence. As shown in Figure 3a, the capsule does not emit fluorescence at a low temperature of 25 °C. When the temperature is increased to 75 °C, the capsule responds rapidly and emits bright green fluorescence with obvious fluorescence color contrast. This result demonstrated that the calcium alginate capsules prepared using the piercing–solidifying method successfully retained the excellent fluorescent properties of the thermochromic fluorescent powders inside. More importantly, the prepared calcium alginate thermochromic fluorescent capsules also exhibited excellent stability in the organic solvent DMAc, as shown in Figure 3b. The thermochromic fluorescent properties of the capsules were retained without obvious change, after a treatment of soaking in DMAc for up to 4 h. This is important for the subsequent process, i.e., wet spinning with PU. The introduction of microencapsulation technology successfully addressed the issue of dye dissolution in organic solvents, ensuring the effective application of the stretchable thermochromic fluorescent fiber. This makes it possible to blend thermochromic fluorescent dyes with PU spinning solution to prepare stretchable fibers with switching fluorescent properties. The reversibility of the elastic thermochromic fluorescent fibers determines the long-term stability of their application and is closely related to the performance of thermochromic fluorescent capsules. To investigate the reversibility of the capsule’s fluorescent performance, it was tested over 50 heating–cooling cycles. As shown in Figure 3c, the capsule still maintains excellent thermochromic fluorescent performance, which fully verifies its good reversibility and stability and provides reliable support for the long-term application of the capsules.

### 3.2. Preparation of Stretchable Thermochromic Fluorescent Fibers

The above-mentioned thermochromic fluorescent capsules were blended with PU spinning solution to prepare elastic thermochromic fibers by wet-spinning technology, as shown in Figure 4a. The PU solution containing thermochromic fluorescent capsules was coagulated by phase inversion, where DMAc was replaced with water. The prepared stretchable thermochromic fluorescent PU fibers were translucent, as observed from Figure 4b. In addition, the thermochromic fluorescent capsules were faintly visible in the fibers. The morphologies of the thermochromic fluorescent PU fibers were characterized by scanning electron microscopy (SEM). As shown in Figure 4c, the fiber diameter is about 765 μm, and the surface has no obvious defects or fractures but is rough with some irregular grooves. This surface morphology is a common feature in wet-spinning technology and is mainly attributed to the substitution between the solvent DMAc in the spinning solution and the deionized water coagulation bath during the spinning process, which leads to the formation of fine undulations and textures on the fiber surface. From the cross-sectional view in Figure 4c, hundreds of micron-sized thermochromic fluorescent capsules were tightly embedded in the fibers, and many thermochromic fluorescent powders were encapsulated inside the capsule. These ensured the temperature-sensitive response properties of the stretchable fibers.

The mechanical properties of the stretchable thermochromic fluorescent fibers were characterized using a single-fiber strength tester at room temperature at a tensile speed of 50 mm/min. In the cyclic tensile tests, the fibers were repeatedly stretched at increasing strain values, as shown in Figure 5. The stretchable fiber can largely recover from large strain values from 100% to 800%. In particular, the strain recovery decreases rapidly at lower applied strains of 100% and 200%, attributing to the irreversible deformation of the molecular chains at the beginning of stretching. The thermochromic fluorescent PU fibers exhibited excellent rebound resilience at several stress levels, even at the high strain of 800%, demonstrating their excellent stretchability. These results demonstrated the application potential of dynamic thermochromic fluorescent PU fibers to the human body.

We further conducted the cyclic loading–unloading tensile test at 100%, 300%, and 500% strain. Figure 6a presents the stress response of the thermochromic fluorescent PU fiber under 100% strain for 100 cycles. It is obvious that the stress level of the fiber tends to a constant value after several cycles, indicating that the fiber has good elasticity and fatigue resistance at this strain level. The stress change of the fiber after 50 consecutive cycles at 300% strain is demonstrated in Figure 6b. The stress level of the fiber increased significantly at 300% strain, compared with that at 100% strain. At this strain level, the fibers may have experienced greater internal structural changes or plastic deformation, but the curves remained somewhat stable after many cycles, showing that the fibers are able to withstand higher strains to some extent. Figure 6c reflects the stress behavior of the fiber after 30 successive cycles at 500% strain. It was observed that the stress level of the fiber increases dramatically as the strain increases further. Compared with that of 100% and 300% strains, the curves in Figure 6c show more pronounced fluctuations and instability during the cycles. This may be attributed to the instability of the mechanical properties of the fibers due to more significant changes and damage to the internal structure of the fibers under high strain levels.

### 3.3. Luminescence Performance of Stretchable Thermochromic Fluorescent Fibers

The thermochromic mechanism of the stretchable fibers based on self-crystallinity phase transition was illustrated, as shown in Figure 7a. Below the phase-change temperature of HcA (64 °C), the molecules of C6 and MB are forced to aggregate in the regular crystalline or semi-crystalline phases of phase-change material HcA. The distance between the fluorophore and quencher is lower than 10 nm, inducing the Förster resonance energy transfer (FRET) [46,47,48] process and significant quenching of the donor’s fluorescence. That is, the stretchable thermochromic fluorescent fiber is in an off-fluorescence state. When heated above the phase-change temperature, the matrix material of the thermochromic fluorescent dye melts and exists in a dispersion state, emitting fluorescent light. It is noted that the fluorescence conversion of the fiber has good reproducibility and high reversibility, which is attributed to the protection of calcium alginate capsules toward the thermochromic fluorescent dyes. In order to evaluate the reversibility of the thermochromic fluorescence of the prepared stretchable fibers, we also performed 50 consecutive heating–cooling cycles. The fluorescence photos of the fibers under different cycle times are shown in Figure 7b, from which it can be observed that the thermochromic fluorescent performance of the fibers remained stable without obvious degradation even after 50 cycles of heating–cooling, which proves that the fibers have good reversibility.

The thermochromic fluorescent PU fibers possess good stability in fluorescent luminescence during the stretching process. As shown in Figure 8, a fiber segment was held by two tweezers and stretched to double, triple, and even quadruple the length of itself. The fiber becomes obviously thinned but still emits bright green fluorescence. The excellent elasticity and luminescence of the stretchable thermochromic fluorescent fiber indicate its good adaptability and huge potential in practical applications on the human body.

## 4. Conclusions

In summary, we have proposed stretchable thermochromic fluorescent fibers based on self-crystallinity phase change, using elastic PU as the fiber matrix, for meeting the dynamic requirements of the human body. The switching fluorescence-emitting characteristic of fibers is derived from the reversible conversion of the dispersion/aggregation state of the fluorophore C6 and the quencher MB in the phase-change material HcA during heating/cooling processes. Considering the important role of phase-change materials, the thermochromic fluorescent dye was encapsuled in the solid state via the piercing–solidifying method to avoid the dissolution of HcA by the organic solvent of the PU spinning solution and maintain excellent thermochromic behavior in the fibers. The fibers obtained by wet spinning exhibited good fluorescent emission contrast and reversibility, as well as high elasticity of 800% elongation. Better luminescent performance and mechanical properties would be achieved by decreasing the size of the thermochromic fluorescent capsules in the future. This work presents a strategy for constructing stretchable smart luminescence fibers for human–machine interaction and communications.

## Figures and Tables

**Figure 1 polymers-16-03575-f001:**
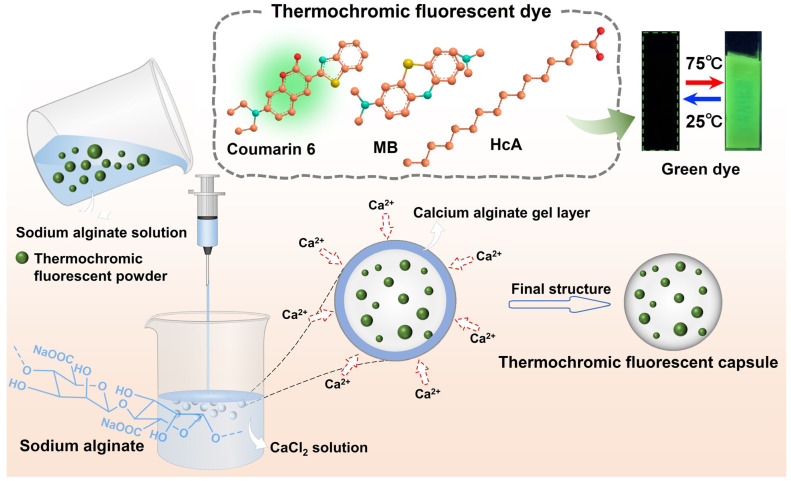
Schematic illustration of the preparation process of thermochromic fluorescent capsules via the piercing–solidifying method.

**Figure 2 polymers-16-03575-f002:**
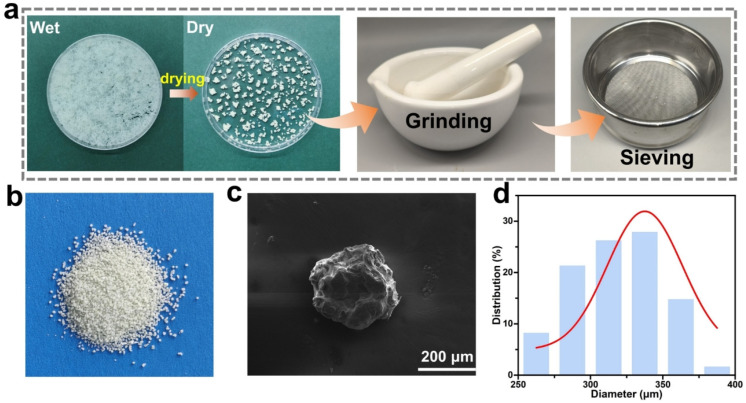
(**a**) Treatment process of the thermochromic fluorescent capsules to be used for wet spinning; (**b**) photograph of the thermochromic fluorescent capsules; (**c**) SEM image of a thermochromic fluorescent capsule; (**d**) size distribution of the thermochromic fluorescent capsules.

**Figure 3 polymers-16-03575-f003:**
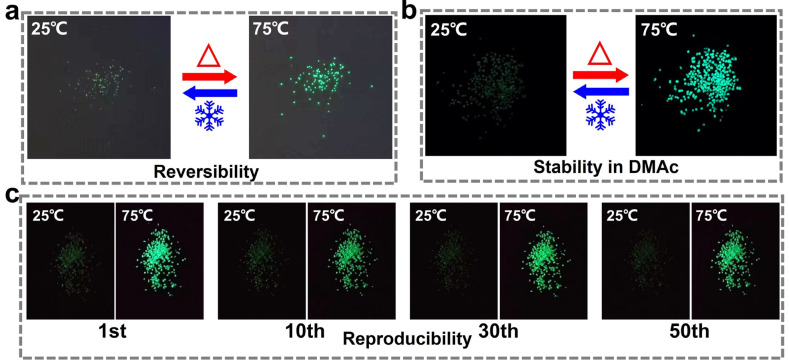
Fluorescent photographs of the thermochromic fluorescent capsules at 25 °C and 75 °C, in (**a**) air and (**b**) DMAc solvent; (**c**) fluorescence switching cycles of the thermochromic capsules at 25 °C and 75 °C.

**Figure 4 polymers-16-03575-f004:**
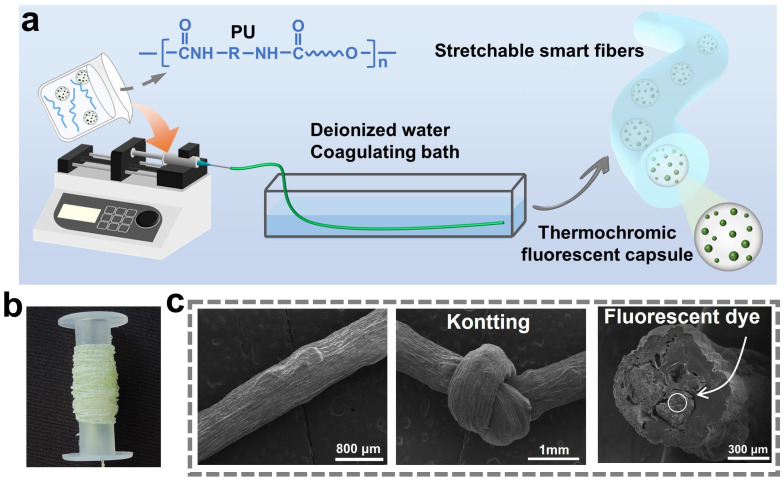
(**a**) Schematic illustration of the wet-spinning process for preparing stretchable thermochromic fluorescent fibers. (**b**) Optical photograph and (**c**) SEM images of the prepared thermochromic fluorescent PU fiber.

**Figure 5 polymers-16-03575-f005:**
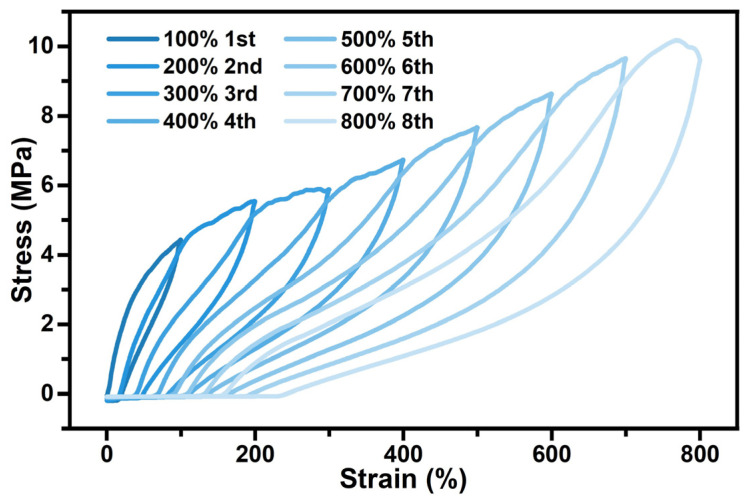
Cyclic loading–unloading tensile curves of the thermochromic fluorescent PU fiber at increasing strain.

**Figure 6 polymers-16-03575-f006:**
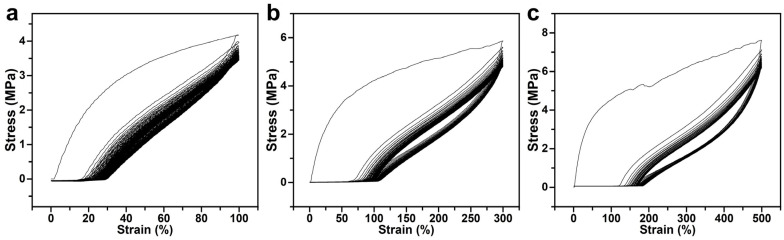
Cyclic loading–unloading tensile curves of the thermochromic fluorescent PU fiber at (**a**) 100% strain, (**b**) 300% strain, and (**c**) 500% strain.

**Figure 7 polymers-16-03575-f007:**
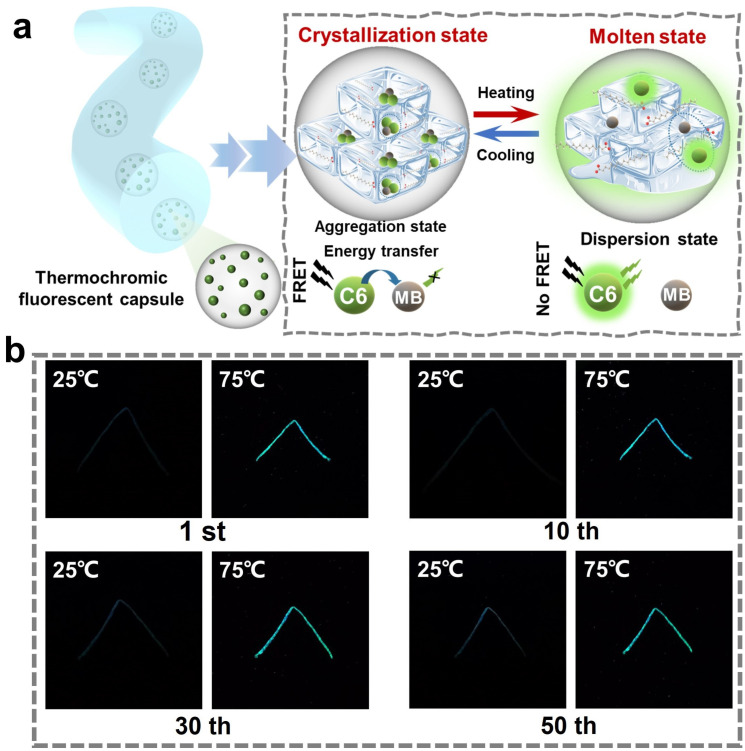
(**a**) Schematic illustration of the thermochromic fluorescent mechanisms of the fibers, based on self-crystallinity phase change. (**b**) Fluorescence switching cycles of the stretchable thermochromic fiber at 25 °C and 75 °C.

**Figure 8 polymers-16-03575-f008:**
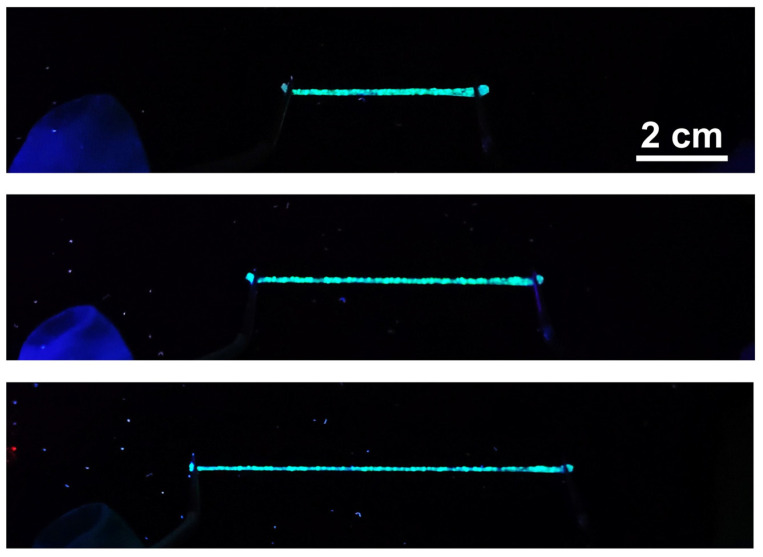
Fluorescent photographs of the thermochromic fluorescent fiber under different stretching levels.

## Data Availability

The original contributions presented in the study are included in the article; further inquiries can be directed to the corresponding author.

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
