# Peer review of "Stretchable Thermochromic Fluorescent Fibers Based on Self-Crystallinity Phase Change for Smart Wearable Displays"

_polymers, 2024, doi:10.3390/polym16243575_

Round 1

Reviewer 1 Report

Comments and Suggestions for Authors

Review of the Manuscript: "Stretchable Thermochromic Fluorescent Fibers Based on Self-Crystallinity Phase Change for Smart Wearable Displays"

General Comments

The manuscript explores a novel and promising area of research focused on smart wearable materials, specifically thermochromic fluorescent fibers with stretchable properties. It effectively demonstrates the preparation and characterization of these fibers using polyurethane (PU) as a matrix and highlights their potential in human-machine interaction and wearable technologies. The work is well-motivated, and the methodology appears thorough and innovative. However, some points require clarification or further discussion to enhance the scientific rigor and clarity.

Major Comments

1.      Innovation and Novelty

The manuscript successfully emphasizes the novelty of integrating self-crystallinity phase-change materials in a stretchable matrix. However, further elaboration is needed on how this approach compares quantitatively to existing methods. For instance:

    • How does the fluorescence contrast and elasticity of these fibers compare to other similar materials?
    • Could the authors provide a table summarizing previous works and their performance metrics?

2.      Material Properties and Behavior

The paper claims an elasticity of up to 800% elongation and stable fluorescence under repetitive cycles. While impressive, the mechanical durability under prolonged cyclic loading should be discussed in more detail. Is there a significant drop in fluorescence or mechanical properties beyond 50 cycles?

3.      Thermochromic Performance

The reversible thermochromic mechanism is central to the study. However, the stability of these properties over extended periods (e.g., months or years) in practical applications remains unclear. A brief discussion or preliminary aging test results would strengthen the argument for real-world applicability.

4.      Applications and Scalability

The manuscript discusses potential applications but lacks specifics regarding the scalability of the production process and real-world implementation challenges. For example:

    • Is the wet spinning method feasible for large-scale manufacturing?
    • Are there any constraints related to the cost or environmental impact of the materials used?

Minor Comments

  1. Clarity in Methodology
    • The description of the piercing-solidifying method is detailed but could benefit from clearer schematics showing the encapsulation steps.
    • Details on the optimization of spinning parameters, such as flow rates or bath conditions, should be included.
  2. Figures and Visuals
    • Figures 3 and 6 are helpful but could benefit from quantitative annotations (e.g., fluorescence intensity values or stress levels).
    • Add a high-resolution image of the final product under typical usage conditions (e.g., integrated into a wearable device).
  3. Language and Grammar
    • Some sections, such as the introduction, contain repetitive phrases and minor grammatical errors (e.g., "a huge challenge is the conformally seamless integration with human body").
    • Revise for conciseness and consistency.
  4. References
    • While the manuscript includes comprehensive references, some key recent works in smart textiles and thermochromic materials might be missing. Cross-check for the most influential papers in the past three years.

Specific Suggestions for Improvement

  1. Clearly highlight the limitations of the current work and propose future directions in the conclusion.
  2. Include statistical analysis (e.g., error bars in the mechanical property tests) to provide confidence in the reproducibility of the results.
  3. Discuss the environmental safety and recyclability of the fibers, particularly given the increasing emphasis on sustainability.

Recommendation

While the manuscript presents a solid foundation, addressing the above points will enhance its impact and clarity. Subject to these revisions, the manuscript is suitable for publication in Polymers.

Comments on the Quality of English Language

Some sections, such as the introduction, contain repetitive phrases and minor grammatical errors (e.g., "a huge challenge is the conformally seamless integration with human body").

Author Response

Dear Ms. Gabriela Bochenek,

Thank you very much for your efficient work on our manuscript entitled as " Stretchable Thermochromic Fluorescent Fibers Based on Self-Crystallinity Phase Change for Smart Wearable Displays" (Manuscript ID: polymers-3356201). Herein, we submit our revised manuscript after carefully taking into the consideration of reviewers’ comments and making the corresponding modifications. Changes requested by the reviewers are indicated in yellow in the marked copy of the manuscript.

We are looking forward to a positive decision concerning its publication in Polymers.

Sincerely yours,

Ying MA

Email: yingma@dhu.edu.cn

************

Response to the reviewers’ comments: (Remarks of the reviewers are in italics and our responses to the remarks are in blue color.)

Reviewer 1:

Grateful for your positive evaluation and professional remarks to our work. We would like to thank reviewer 1 for taking the time for carefully reading our manuscript and for providing us with detailed and important suggestions that were very helpful for improving the manuscript.

C1: How does the fluorescence contrast and elasticity of these fibers compare to other similar materials? Could the authors provide a table summarizing previous works and their performance metrics?

Response: Thanks for the reviewer's suggestion. The thermochromic fluorescent dyes based on self-crystallinity phase change and Förster resonance energy transfer possess good fluorescent emission contrast and reversibility, which have been proven by Zhang et. al. (Adv. Mater. 2021, 33, 2008055). After being mixed in fibers, the fluorescence intensity of thermochromic fluorescent powders decreases slightly due to the presence of polymer matrix. The values of fluorescent emission contrast remained around ~5 in our previous work at Adv. Opt. Mater. Accepted), which is obviously higher than the brightness discrimination threshold of naked eyes (0.005~0.02). This work is about the electrochromic fluorescent fiber prepared by the same thermochromic fluorescent dyes. However, we did not measure the fluorescence spectrum of thermochromic fluorescent fibers in this manuscript, due to the difficulty in controlling the fiber temperature in a fluorescence spectrophotometer. 

Moreover, the polymer matrix in this Manuscript is the commercial polyurethane fibers, due to the excellent elasticity. The elasticity of the prepared thermochromic fluorescent fiber have been tested and presented in Figure 5 and 6. This presentation is classic and have been used in many reported works.

C2: Is there a significant drop in fluorescence or mechanical properties beyond 50 cycles?

Response: Thanks to the reviewer for the important question. The stability is very important. However, we did not measure the fluorescence spectrum of thermochromic fluorescent fibers in this manuscript, due to the difficulty in controlling the fiber temperature in a fluorescence spectrophotometer. But we totally agree with you and demonstrated that in our previous work (Adv. Opt. Mater. Accepted). An electrochromic fluorescent fiber was prepared by coaxial wet-spinning, and the color was controlled by the inner electrical heating wire. The electric control is easier than thermal, especially in a fluorescence spectrophotometer. As shown in the figure below, the fluorescence of the fiber is repeatable during 50 cycles, but with a slight decrease. The fluorescence decrease is possibly attributed to the diffusion of dye particles at the interfaces of the polymer matrix. Thus, we considered that the luminous performance of the thermochromic fluorescent fibers in this work have good stability, due to the introduction of capsules. Moreover, the stability of the mechanical properties is also good, due to the polymer matrix of fiber is commercial PU.

C3: The reversible thermochromic mechanism is central to the study. However, the stability of these properties over extended periods (e.g., months or years) in practical applications remains unclear. A brief discussion or preliminary aging test results would strengthen the argument for real-world applicability.

Response: Thanks to the reviewer. From by eyes, the stability is good after one month. However, we did not measure the fluorescence spectrum of thermochromic fluorescent fibers in this manuscript, due to the difficulty in controlling the fiber temperature in a fluorescence spectrophotometer.

C4: Is the wet spinning method feasible for large-scale manufacturing? Are there any constraints related to the cost or environmental impact of the materials used?

Response: Thanks to the reviewer. Wet spinning is a relatively simple and was used as an example of spinning method to prepare the smart fiber. Of course, there are many types of polymers or hydrogels that have been shown in literature to create fibers, either wet spinning, electrospinning, and even melt spinning at a low temperature.

C5: The description of the piercing-solidifying method is detailed but could benefit from clearer schematics showing the encapsulation steps. Details on the optimization of spinning parameters, such as flow rates or bath conditions, should be included.

Response: Thanks to the reviewer. More details of spinning parameters, such as flow rates or bath conditions, have been added in experimental section in Page 7 of the revised Manuscript. “The spinning solution was then fed into a 12 G spinneret (with an inner diameter of 2.3 mm) through a syringe pump at a flow rate of 1 mL/min. Simultaneously, the solution was drawn into a deionized water coagulation bath at a rate 1.5 times faster than the feed rate.”

C6: Figures 3 and 6 are helpful but could benefit from quantitative annotations (e.g., fluorescence intensity values or stress levels). Add a high-resolution image of the final product under typical usage conditions (e.g., integrated into a wearable device).

Response: Thanks to the reviewer. We can not measure the fluorescence spectrum of thermochromic fluorescent fibers in this manuscript, due to the difficulty in controlling the fiber temperature in a fluorescence spectrophotometer. According to the stability of the electrochromic fluorescent fiber prepared by same thermochromic fluorescent dyes in our previous work (Adv. Opt. Mater. Accepted), we considered that the luminous performance of the thermochromic fluorescent fibers in this work have good stability, due to the introduction of capsules.

The polymer matrix in this Manuscript is the commercial polyurethane fibers. The elasticity of the prepared thermochromic fluorescent fiber have been tested and presented in Figure 5 and 6. This presentation is classic and have been used in many reported works. Moreover, the integration of the elastic fiber will be discussed in the next work, on the assist of AI technique.

C7: Some sections, such as the introduction, contain repetitive phrases and minor grammatical errors (e.g., "a huge challenge is the conformally seamless integration with human body"). Revise for conciseness and consistency.

Response: Thanks for the reviewer's suggestion. We have revised the sentence as shown as follows.

In Page 3: “However, a huge challenge of smart fibers as one of wearable materials is stretchable capability for seamless integration with human body.”

In Page 4: “However, for wearable materials, a huge challenge is the seamless integration with human body, so it is particularly important to design smart fibers with stretchable capability to adaptive the dynamic deformation on human body.”

In Page 5: “Despite these efforts to be devoted, it remains a significant challenge for the smart fluorescent fibers based on self-crystallinity phase change to produce the stretchable capability for seamless integration with human body.”

C8: While the manuscript includes comprehensive references, some key recent works in smart textiles and thermochromic materials might be missing. Cross-check for the most influential papers in the past three years.

Response: Thanks to the reviewer. As suggested, some highly relevant literatures on smart textiles and thermochromic materials have been cited in Reference of the revised Manuscript.

C9: Clearly highlight the limitations of the current work and propose future directions in the conclusion.

Response: Thanks to the reviewer. As the suggested, we have added “The better luminescent performance and mechanical properties would be achieved by decreasing the size of thermochromic fluorescent capsules in the future.” in Page 18 of the revised Manuscript.

C10: Include statistical analysis (e.g., error bars in the mechanical property tests) to provide confidence in the reproducibility of the results.

Response: Thanks to the reviewer. We have shown the statistical analysis of multiple cyclic stretching in Figure 6, consistent with the properties of polyurethane fibers during stretching.

The elasticity of the prepared thermochromic fluorescent fiber have been tested by a classic method and presented in Figure 5 and 6. The curves is real-time test, which is different with the strain-stress curve.

C11: Discuss the environmental safety and recyclability of the fibers, particularly given the increasing emphasis on sustainability.

Response: Thanks to the reviewer. The environmental safety and recyclability is not in the study scope of this work.

Reviewer 2:

Grateful for your positive evaluation to the novelty of our work. We would like to thank reviewer 2 for taking the time for carefully reading our manuscript and for providing us with detailed and important suggestions that were very helpful for improving the manuscript.

C1: Details on PU formulation should be provided to better understand the PU properties measured.

Response: Thanks to the reviewer. Polyurethane (PU) was obtained from Huntsman Polyurethanes Co., Ltd. We used polyurethane as a flexible spinning substrate to provide good elasticity and mechanical properties for Stretchable thermochromic fluorescent fibers.

C2: There is no thermal analysis. This is the main drawback of this manuscript. In my opinion it is essential to understand the role of the PCM. For instance, which is the melting point? Is there any kind of chemical interaction between components of the thermochromic particle?

Response: Thanks to the reviewer. The melting point of PCM is 64℃, which have detailed describe in our previous works.( ACS Appl. Mater. Interfaces 2021, 13, 57943; Small 2024, 2310762; Adv. Opt. Mater. Accepted) As suggested, we added the description in Page 15 in the revised Manuscript. “Below the phase-change temperature of HcA (64°C), the molecules of C6 and MB are forced to aggregate in the regular crystalline or semi-crystalline phases of phase-material HcA.”

The thermochromic mechanism of the stretchable fibers based on self-crystallizing phase transition was presented in Figure 7a Page 15 of the manuscript. “Below the phase-change temperature, the molecules of C6 and MB are forced to aggregate in the regular crystalline or semi-crystalline phases of phase-material HcA. The distance between the fluorophore and quencher is lower than 10 nm, inducing Förster resonance energy transfer (FRET) process and significant quenching of the donor’s fluorescence. That is, the stretchable thermochromic fluorescent fiber is at off-fluorescence state. When heated above the phase-change temperature, the matrix material of thermochromic fluorescent dye melts and exists at the dispersion state, emitting the fluorescent light. It is noted that the fluorescence conversion of fiber has good reproducibility and high reversibility, which is attributed to the protection of calcium alginate capsules for the thermochromic fluorescent dyes.”

C3: To present mechanical results in this way, dimensions of the fibers should be provided., as well as replicates among different fibers.

Response: Thanks to the reviewer. The mechanical curves are designed to show that thermochromic fluorescent fibers have good elasticity. The elasticity of the prepared thermochromic fluorescent fiber have been tested by a classic method and presented in Figure 5 and 6. The curves is real-time test to show the elasticity.

C4: From the SEM image it can be seen that the relative size of the microcapsule related to the fiber section is important and may strongly affect its mechanical resistance. There is no analysis on the failure mechanism.

Response: Thanks to the reviewer. Indeed, the interfacial interactions between the microcapsules and the polyurethane matrix decreased mechanical properties. If the microcapsules are not compatible with the polyurethane matrix, voids or cracks may develop at the interface, which may reduce the overall strength of the material. Due to the big difference among the capsule size, we only tested the elasticity of fiber.

Reviewer 2 Report

Comments and Suggestions for Authors

The manuscript presents an interesting concept. However some aspects should be considered an discussed before its publication.

1.- Details on PU formulation should be provided to better understand the PU properties measured.

2. There is no thermal analysis. This is the main drawback of this manuscript. In my opinion it is essential to understand the role of the PCM. For instance, which is the melting point? Is there any kind of chemical interaction between components of the thermocromic particle?

3. To present mechanical results in this way, dimensions of the fibres should be provided., as well as replicates among different fibres.

4. From the SEM image it can be seen that the relative size of the microcapsule related to the fibre section is important and may strongly affect its mechanical resistance. There is no analysis on the failure mechanism.  

Author Response

(The authors gave the same response as above.)

Round 2

Reviewer 1 Report

Comments and Suggestions for Authors

The manuscript can be accepted as it is.